# Development of Antimicrobial Surfaces Using Diamond-like Carbon or Diamond-like Carbon-Based Coatings

**DOI:** 10.3390/ijms25168593

**Published:** 2024-08-06

**Authors:** Yasuhiro Fujii, Tatsuyuki Nakatani, Daiki Ousaka, Susumu Oozawa, Yasushi Sasai, Shingo Kasahara

**Affiliations:** 1Center for Innovative Clinical Medicine, Okayama University Hospital, Okayama University, Okayama 700-8558, Japan; 2Institute of Frontier Science and Technology, Okayama University of Science, Okayama 700-0005, Japan; nakatani@ous.ac.jp; 3Department of Pharmacology, Faculty of Medicine, Dentistry and Pharmaceutical Sciences, Okayama University, Okayama 700-8558, Japan; ousaka@md.okayama-u.ac.jp; 4Division of Medical Safety Management, Safety Management Facility, Okayama University Hospital, Okayama University, Okayama 700-8558, Japan; ohzawa-s@cc.okayama-u.ac.jp; 5Department of Pharmacy, Gifu University of Medical Science, Kani 509-0293, Japan; ysasai@u-gifu-ms.ac.jp; 6Department of Cardiovascular Surgery, Faculty of Medicine, Dentistry and Pharmaceutical Sciences, Okayama University, Okayama 700-8558, Japan; shingok@md.okayama-u.ac.jp

**Keywords:** diamond-like carbon, antibacterial surface, hydrophilicity, ζ-potential, surface smoothness, biofilm, bacterial adhesion

## Abstract

The medical device market is a high-growth sector expected to sustain an annual growth rate of over 5%, even in developed countries. Daily, numerous patients have medical devices implanted or inserted within their bodies. While medical devices have significantly improved patient outcomes, as foreign objects, their wider use can lead to an increase in device-related infections, thereby imposing a burden on healthcare systems. Multiple materials with significant societal impact have evolved over time: the 19th century was the age of iron, the 20th century was dominated by silicon, and the 21st century is often referred to as the era of carbon. In particular, the development of nanocarbon materials and their potential applications in medicine are being explored, although the scope of these applications remains limited. Technological innovations in carbon materials are remarkable, and their application in medicine is expected to advance greatly. For example, diamond-like carbon (DLC) has garnered considerable attention for the development of antimicrobial surfaces. Both DLC itself and its derivatives have been reported to exhibit anti-microbial properties. This review discusses the current state of DLC-based antimicrobial surface development.

## 1. Introduction

The global medical devices market was valued at United States dollar (USD) 570 billion in 2022 and is expected to reach around USD 996.93 billion by 2032, poised to expand at a compound annual growth rate of 5.8% during the 2023–2032 forecast period [1]. However, the increasing use of medical devices will lead to a rise in device-associated infections, which will, in turn, impose a burden on healthcare systems. Therefore, medical device-associated infections represent an urgent issue. The antimicrobial treatment of materials used in medical devices represents one of the crucial solutions to this problem.

The prominence of materials shifts with time. The 19th century was defined by iron, the 20th by silicon, and the 21st century is hailed as the century of carbon. Humanity has used carbon materials for ages, with diamond and graphite being the most notable examples. Carbon fibers were developed in the early 1960s and were initially used in rockets, accelerating the development of modern carbon materials. Currently, notable carbon materials include diamond (diamond structure sp^3^), graphene (a sheet of graphite structure sp^2^), fullerene (a structure where carbon atoms are arranged according to Euler’s polyhedron theorem), carbon nanotubes (graphene sheets that form single or multi-layered co-axial tubes), and diamond-like carbon (DLC, a combination of sp^3^ and sp^2^ structures that may also contain hydrogen). These materials have garnered significant attention because they enable properties and applications that were previously unattainable with traditional metal and organic materials.

DLC was first synthesized around 1970 by Aisenberg et al. using ion beam deposition [2]. Initially, it attracted attention as a tribological material due to its low friction coefficient [3], leading to its gradual adoption in industrial applications. In the medical field, the high biocompatibility of DLC became an area of interest around 1990 [4], further expanding its range of applications. Since 2000, there has been an increase in reports on the bio-medical potential of DLC. Efforts have been made to impart biocompatibility to medical devices [5,6], provide antibacterial properties [7,8], and control cell adhesion as well as the endothelialization of foreign surfaces [6,9]. Additionally, new coating materials based on DLC have been increasingly reported, including those doped with oxygen [10,11], nitrogen [12,13], metal [14,15], and halogen atoms [16,17], as well as those fused with polymer coating materials [18,19]. Consequently, DLC has gained attention as a promising bio-medical coating material for implantable medical devices.

DLC and DLC-based coatings hold significant potential for enhancing the antibacterial properties of medical devices in the future. However, the field of medical carbon coatings is still in its infancy, and their efficacy is not yet widely recognized among healthcare professionals. In this review, we discuss the interfacial properties that should be considered when antibacterial surface properties are required for medical devices. We also explore the potential antibacterial performance that can be achieved through DLC and DLC-based coatings. To the best of our knowledge, this review represents the first comprehensive summary of diamond-like carbon (DLC) and DLC-based antimicrobial coatings.

For this review, literature searches were conducted using PubMed and J-STAGE. All relevant literature up to the end of April 2024 was included in the search. The search term “Diamond-like carbon” was used to identify relevant documents, focusing on both reviews related to DLC and research articles addressing the antimicrobial properties of DLC. This approach allowed us to collect comprehensive information on the subject.

## 2. Requirements for Conferring Antibacterial Properties to Object Surfaces

To establish a bacterial infection on a foreign material interface, the following sequential steps are necessary (Figure 1):Landing: Bacteria initially land on the surface of the foreign material.Adhesion and aggregation: Bacteria adhere to the surface and begin to aggregate.Biofilm formation: Bacteria protect themselves by forming a biofilm, creating small colonies.Colony maturation: The bacterial colonies mature, enhancing their resilience.Dispersal: Bacteria disperse from the mature colonies to colonize new areas.

By targeting and inhibiting these steps, the surfaces of foreign materials can be endowed with antibacterial properties. Specifically, this can be achieved by:Preventing bacterial landing: Implementing strategies to keep bacteria from initially landing on the surface.Inhibiting adhesion and aggregation: Developing methods to prevent bacteria from adhering to and aggregating on the surface.Suppressing biofilm formation: Employing techniques to inhibit biofilm formation, thereby preventing the formation and maturation of bacterial colonies.

Achieving one or more of these goals can impart antibacterial properties to the foreign material interface. Understanding the specific requirements for the surface of the foreign material is crucial for designing effective antibacterial surfaces.

This review aims to organize and present the knowledge necessary for designing such antibacterial surfaces. By utilizing diamond-like carbon (DLC) and DLC-based technologies, it is possible to modify the following surface properties:Free energy;Tribology (smoothness, lubricity, friction, and wear);Topography;Surface chemical characteristics;Surface electrical properties;Surface Elasticity.

These surface properties can be controlled to a certain extent when using DLC as a single base material. It is believed that changes in these surface properties will also affect antimicrobial performance.

## 3. Surface Free Energy

Similarly to the surface tension in liquids caused by intermolecular forces, solids also experience a force that acts to minimize their surface area due to intermolecular forces. This force in solids is referred to as “surface free energy”. It can be quantitatively assessed by the contact angle with water; a larger contact angle indicates lower surface free energy (hydrophobicity), whereas a smaller contact angle signifies higher surface free energy (hydrophilicity). In the medical field, the terms hydrophilicity and hydrophobicity are commonly used when describing these properties. The following discussion will thus explore the relationship between antimicrobial properties and hydrophilicity/hydrophobicity.

Some studies have reported on the antimicrobial properties of materials with extreme hydrophobicity or hydrophilicity. Superhydrophilic graphene oxide [21], superhydrophilic polyvinylidene fluoride membranes containing titanium dioxide nanoparticles [22], superhydrophobic titanium dioxide-coated cellulose fiber substrates [23], and superhydrophobic Ag/TiO_2_ nanotubes [24] have all been described. The antimicrobial properties of these materials are likely due to a combination of factors rather than just their hydrophobicity or hydrophilicity. However, some studies have suggested that highly hydrophilic coatings can reduce bacterial adhesion [25], whereas superhydrophobic surfaces with reduced solid area fractions can enhance self-cleaning abilities, making the removal of bacteria with water easier [26]. Therefore, extreme hydrophilicity or hydrophobicity is expected to improve antimicrobial properties. On the other hand, bacteria tend to adhere more easily to surfaces with moderate hydrophilicity than to those with superhydrophilic or superhydrophobic properties [27,28]. It has been reported that cell adhesion is enhanced at a contact angle of about 55° [29] because bacterial peptidoglycan adsorbs more easily at contact angles between 54° and 130° [30]. Further, *E. coli* adhesion is maximized on polymeric surfaces with a contact angle of 90° [26]. The specific water contact angle at which bacterial adhesion is maximized is likely to vary depending on the substrate, type of DLC, surrounding environment, and bacterial species. Therefore, a definitive threshold will be case-dependent. However, the observation that bacterial adhesion is most significant at intermediate water contact angles is an important finding.

The water contact angle varies depending on the coating substrate and the type of DLC, but the water contact angle of DLC itself ranges from approximately 70° to 100° [5,8,11,19,31]. From the perspective of surface free energy, this does not significantly contribute to antimicrobial properties. However, in one of our studies, DLC alone was reported to exhibit antimicrobial activity against Staphylococcus aureus on polyurethane substrates. This is a comprehensive result of the increased smoothness provided by the DLC, the change in water contact angle from 90° to 74.5°, and the shift in zeta(ζ)-potential measured in a 10 mM NaCl environment from approximately −4 mV to around −11 mV [7]. Similar activity has also been described against Pseudomonas aeruginosa and Escherichia coli in artificial urine environments within silicone tubes. The inner lumen of silicone tubes was coated with DLC, and artificial urine containing bacteria was circulated through these tubes for 14 days. The DLC coating inhibited the colonization of Pseudomonas aeruginosa and Escherichia coli on the inner surface of the silicone tubes and significantly reduced biofilm formation by Pseudomonas aeruginosa [8]. However, in this study, no antimicrobial effect against Staphylococcus aureus was observed under these conditions, suggesting that the antimicrobial performance of DLC may vary depending on the application context. Furthermore, doping DLC with fluorine significantly increases its hydrophobicity, whereas doping with oxygen or nitrogen greatly enhances hydrophilicity. These modifications are relatively straightforward and significantly alter the surface free energy of DLC, making it a suitable coating material for controlling surface energy. Our ongoing research on DLC coatings for resins has shown that, while resins typically exhibit poor polymer adhesion, using DLC as a substrate improves polymer adhesion. By coating a resin substrate with a 2-methacryloyloxyethyl phosphorylcholine (MPC) polymer brush over a DLC base, we successfully created a durable MPC polymer coating, resulting in an ultra-hydrophilic surface. This demonstrates the potential of hybrid surfaces combining DLC with polymer coatings for advanced surface control. Figure 2 shows examples of contact angle changes obtained via processing DLC.

## 4. Tribology

The high smoothness, low friction coefficient, and high wear resistance of DLC have been recognized since its inception. In the automotive industry, where improving fuel efficiency by reducing friction losses is a critical objective, the application of DLC rapidly expanded in the early 2000s [32]. We have conducted research on DLC (diamond-like carbon) coatings on resin substrates and confirmed that DLC coatings enhance surface smoothness on resins similarly to metals. In ePTFE, DLC coating resulted in a 65% reduction in arithmetic mean roughness (Ra) and a 68% reduction in root mean square roughness (Rq) [5]. Similarly, for polyurethane substrates, DLC coating reduced the surface Ra by 70% and Rq by 45% [7], and on silicon surfaces, DLC coating decreased Ra by 70% [8]. Although these values were likely influenced by the original surface microstructure and slight compositional differences of the substrates used, it is believed that the smoothness enhancement effect of DLC coating is also applicable to resin materials. Surface roughness and texture are also considered crucial for the antimicrobial performance of substrate surfaces [33]. Increased smoothness is associated with low friction [34], whereas a larger surface area provides more area for bacterial adherence, eventually supporting bacterial growth [30,35,36,37]. However, in practical applications, the relationship between air and surface roughness must also be considered. Superhydrophobic surfaces are created using surface roughness, which traps air [38], reducing the contact area between bacteria and the surface, thus making bacterial adhesion more difficult [39,40,41,42,43]. However, in the absence of air, rough surfaces exhibit a greater surface area, which facilitates bacterial adhesion [30,35,36,37]. When DLC is coated onto fibrous structures such as extended polytetrafluoroethylene (ePTFE), each microfiber can be observed as uniformly covered with the DLC coating (Figure 3). In such cases, changes in topographic patterns are expected to induce tribological changes. It is generally accepted that large surface areas with rough textures promote bacterial adhesion. However, due to the difficulty of establishing conditions that vary only in smoothness, quantifying the extent to which smoothness affects bacterial activity or identifying specific topographic patterns that promote adhesion remains challenging. It is hypothesized that the size and shape of the bacteria also influence this phenomenon [44,45,46,47,48,49,50].

## 5. Surface Chemical Composition and Hardness

Figure 4 presents a classification of DLC based on sp^2^ bonds, sp^3^ bonds, and hydrogen content, alongside the hardness of 74 different types of DLC (with larger circles indicating greater hardness) [31]. DLC exhibits a hybrid structure comprising sp^2^ carbon bonds formed by σ bonds and sp^3^ carbon bonds formed by π bonds. DLC can be categorized into four types based on the hydrogen content and the dominant hybridization: a-C (amorphous carbon, primarily sp^2^ hybridized with minimal hydrogen content), ta-C (tetrahedral amorphous carbon, primarily sp^3^ hybridized with minimal hydrogen content), a-C:H (amorphous hydrogenated carbon, primarily sp^2^ hybridized with hydrogen), and ta-C:H (tetrahedral amorphous hydrogenated carbon, primarily sp^3^ hybridized with hydrogen). The hydrogen content significantly influences the hardness of DLC, with a higher hydrogen content resulting in a softer material. When the hydrogen content is substantial, the material is termed polymer-like carbon (PLC), which is distinct from DLC, though the boundary between these two is not clearly defined [31,51] (Figure 4). Hydrogen has been reported to exhibit antibacterial properties. This antimicrobial effect is primarily attributed to the generation of reactive oxygen species (ROS), particularly hydrogen peroxide (H₂O₂), which is well-documented for its potent antimicrobial activity. The ingestion of hydrogen-rich water and inhalation of hydrogen gas have been shown to reduce oxidative stress and suppress inflammatory responses in the body, thereby improving the internal environment and potentially enhancing immune function, which may indirectly inhibit bacterial infections [52]. These reactions are unlikely to occur when hydrogen and carbon are tightly chemically bonded, as in diamond-like carbon (DLC). However, the process of growing hydrogen-containing DLC is highly complex [53,54], leading to variability in the modes of hydrogen presence. Moreover, within DLC, hydrogen exists in two forms: hydrogen that is tightly chemically bonded to carbon and hydrogen that is present but not tightly bonded, effectively trapped within the carbon matrix. It is suggested that there are regions within the DLC where trapped hydrogen accumulates, resulting in variations in hydrogen concentration [55]. While loosely bound hydrogen is speculated to have potential antimicrobial activity, no studies have been found to verify this. Detailed research is needed on the specifics of hydrogen presence, the ratio of tightly bonded to loosely bonded hydrogen, and the biological activity of loosely bonded hydrogen. Nevertheless, the DLC developed for our resin interior has a high hydrogen content of 30%, and under certain conditions, we have confirmed its antimicrobial properties [7,8]. It is intriguing to consider whether the hydrogen within the DLC contributes to this effect. Additionally, several studies have reported the antimicrobial properties of DLC films. Levon et al. reported that DLC induced lower adherence and colony formation by *Staphylococcus aureus* than tantalum, titanium, and chromium [56]. Del Prado et al. also demonstrated the antibacterial effectiveness of DLC-coated ultra-high-molecular-weight polyethylene against nine different staphylococcal species [57]. Further research is required in order to elucidate the relationship between the structure of DLC and its antibacterial properties.

In the processing of carbon films, it is possible to dope the material with various atoms. For example, by performing a brief surface treatment with oxygen plasma for a few to tens of seconds after coating DLC, the oxygen content within DLC increases significantly, thereby altering its properties. Doping atoms into DLC can enhance its functionality as a biological material. The process of doping atoms into DLC is relatively straightforward and allows for adjustment of the dopant atom concentration. There are numerous reports on the antibacterial properties imparted to DLC through atomic doping. Antibacterial performance has been demonstrated with the doping of oxygen (O) [11], silver (Ag) [14,58,59,60,61,62], copper (Cu) [14,62,63], titanium (Ti) [64], fluorine (F) [65], chromium (Cr) [66], zinc (Zn) [67], and germanium (Ge) [68]. These studies indicate that DLC’s antibacterial capabilities can be significantly enhanced through careful selection and incorporation of different dopant atoms. However, the antibacterial mechanism observed when atoms are doped is primarily due to the cytotoxicity of the doped atoms. Therefore, careful consideration of the side effects related to cytotoxicity is necessary before practical application. Notably, many metal atoms exhibit strong cytotoxicity when ionized [14,67]. Additionally, unexpected side effects that cause serious harm to biological systems may also arise beyond cytotoxicity. For instance, we observed that oxygen doping significantly reduces the blood compatibility of DLC surfaces (unpublished data), which were originally comparable to that of ePTFE vascular grafts [5]. Consequently, oxygen-doped DLC cannot be used for blood-contacting applications such as intravascular stents, artificial blood vessels, or vascular catheters. It is important to always bear in mind that doping atoms can significantly alter the properties of DLC.

## 6. Surface Electrical Properties

Surface potential is a crucial factor in understanding the interactions between bacteria and substrates. Generally, bacterial cells are covered with peptidoglycan, composed of sugars and amino acids, which results in most bacterial surfaces being negatively charged. Consequently, positively charged surfaces are employed to capture or kill bacteria, whereas negatively charged surfaces are used to repel bacteria with polyanionic glycosides, which are predominantly gram-positive [69]. However, the situation is not straightforward, as some gram-positive cocci can adhere to negatively charged surfaces [70]. Notably, surfaces with both cationic and anionic functional groups, known as zwitterionic surfaces, have been reported to strongly inhibit protein adsorption and bacterial adhesion [71,72,73]. Moreover, the mechanisms underlying this phenomenon emphasize two key points: (1) zwitterionic surfaces with strong ionic solvation can bind water molecules strongly and stably through electrostatic interactions, forming a hydration layer that acts as an energetic and physical barrier to resist protein adsorption; (2) the neutrally charged zwitterionic surface can minimize electrostatic interactions between proteins/bacteria and the material [74]. DLC holds considerable potential for the fabrication of such zwitterionic surfaces. By doping with oxygen and nitrogen, it is possible to create surfaces with zwitterionic functional groups, and attaching zwitterionic substances to DLC may enhance its performance. Lee et al. reported that coating titanium–nickel vascular stents with DLC and coupling zwitterion (N^+^ and SO_3_^−^)-linked poly(ethylene glycol) (PEG) reduced protein adsorption and improved stent biocompatibility due to its antifouling effects [75].

In addition, the ζ-potential serves as an indicator of the charge states of solid surfaces in solution. A material is surrounded by ions with a potential opposite to that of its surface, forming an ion-fixed layer on its surface. Beyond this fixed layer, an ion diffusion layer forms, containing ions with potential opposite to that of the fixed layer. Under an electric field, a boundary is established between ions that move independently of the material (within the ion diffusion layer) and those that move with the material. This boundary is referred to as the “slipping plane,” and the potential at this plane is known as the ζ-potential (Figure 5). Unlike the surface potential, the ζ-potential varies with the surrounding environment, including the types and concentrations of ions in solution, the temperature, and the pH of the solution. While we will not discuss the relationship and measurement methods of surface potential and ζ-potential in detail here, it is notable that ζ-potential is relatively stable and easier to measure than surface potential. Moreover, in biological reactions, the ζ-potential is crucial as it influences substances and cells in the solution before they reach the surface, highlighting its importance in bio-response studies. Using a proprietary resin inner lumen DLC coating method, we applied a DLC coating to a polyurethane sheet and observed a change in ζ-potential from approximately −4 mV to approximately −10 mV. This demonstrated that the DLC coating effectively inhibited the adhesion of *Staphylococcus aureus* to the polyurethane. Although this comprehensive change can be attributed to improvements in smoothness and alterations in hydrophilicity, it is logical that the increased negative ζ-potential further reduces the adhesion of *S. aureus*, given its reported net charge of −35.6 mV [76]. The relationship between ζ-potential and antibacterial properties is a crucial factor in designing antimicrobial surfaces with DLC; however, current knowledge on this matter remains limited.

Theoretically, a completely DLC-coated surface should exhibit a consistent ζ-potential value regardless of the substrate because ζ-potential is considered an intrinsic property of materials. However, we currently observe variations in ζ-potential depending on the substrate. We hypothesize that this is due to the presence of pinholes in the DLC coating, though small in area, through which the substrate influences the ζ-potential of the DLC surface. Since DLC is deposited via vapor deposition, clusters form in the plasma during deposition, sometimes detaching and leaving gaps that form pinholes. It is particularly likely that thin DLC coatings are more susceptible to the influence of pinholes. As there is currently no technology to completely eliminate pinholes in DLC, variations in potential arise due to differences in substrate material. Moreover, hydrogen-containing DLC is not neutral; the surface of DLC is terminated with hydrogen. Additionally, DLC deposited in a vacuum chamber can incorporate oxygen due to residual oxygen pressure, leading to the formation of carboxyl groups from carbon, hydrogen, and oxygen. Therefore, the ζ-potential of DLC can vary widely. Further research is needed to measure and interpret the ζ-potential of DLC comprehensively.

## 7. Thickness and Density of the Biofunctional DLC Membrane

The thickness and density of DLC films significantly influence their functional properties. Variations in the thickness of DLC films can affect their mechanical properties, wear resistance, and corrosion resistance. For instance, while a thicker film generally provides higher wear resistance, excessively thick films may be prone to delamination. Film density also plays a crucial role. High-density films are typically harder and exhibit superior chemical resistance, whereas lower density can compromise the film’s consistency and protective capabilities. The characteristics of DLC films are also influenced by the manufacturing techniques and process conditions employed. Thus, selecting the appropriate film thickness and density is crucial for achieving the desired properties for specific applications. Although there appears to be a lack of literature directly examining the correlation between antibacterial properties and DLC films, it is evident that film stability impacts the functionality and durability of DLC. Generally, the film thickness increases with longer deposition times. In our study, we coated ePTFE substrates with DLC films of 20, 100, and 300 nm thicknesses by varying the deposition time and conducted tensile tests. Our results indicated that thicker DLC films tend to cause the fibrous structure of ePTFE to fracture more easily.

## 8. Surface Diffusion

Finally, I would like to introduce the concept of surface diffusion, which may be unfamiliar in the field of medicine. Surface diffusion refers to the movement of atoms, molecules, and atomic clusters on a solid surface [77]. This phenomenon significantly impacts the properties and behavior of materials, influenced by factors such as temperature and surface energy states. Surface diffusion is known to profoundly affect the kinetics of adsorption, desorption, and reactions on surfaces [78]. To the best of our knowledge, there have been no reports examining the relationship between antimicrobial effects and surface diffusion. However, given its impact on surface adsorption and related processes, it is anticipated that surface diffusion could also substantially influence antimicrobial performance. DLC is generally known for its high hardness and chemical stability, making surface diffusion less likely to occur. However, the incorporation of hydrogen, oxygen, or other dopants can influence the mechanical and chemical properties of DLC, potentially promoting the movement of surface atoms. While further research is necessary to elucidate the specific behaviors and effects of surface diffusion in DLC, it is essential to understand this phenomenon in order to optimize the application of DLC in biomedical materials.

## 9. Biocompatibility

Diamond-like carbon (DLC) has been reported as a superior coating material in terms of biocompatibility. Hemocompatibility is one of the most demanding aspects of biocompatibility, and our DLC developed for the inner lumen of resin materials has demonstrated hemocompatibility comparable to that of ePTFE, **the current** clinical standard for hemocompatible biomaterials, in multiple animal and human whole-blood contact tests [5]. This property may vary depending on the type of DLC. Additionally, DLC has been reported to aid in the endothelialization of medical devices, thus preventing thrombosis [6]. This antithrombotic effect is believed to result both directly **from** the properties of DLC and indirectly from its influence on **cellular dynamics**. Basic research aimed at the clinical application of DLC-based coatings has explored their use in vascular stents [79], artificial valves [80], and guide wires [81]. Particularly, clinical trials have been conducted on cobalt–chromium coronary stents coated with DLC, showing superior outcomes compared to bare metal stents [82]. DLC-coated coronary stents competed for the position of next-generation stents against drug-eluting coronary stents. However, the trials were discontinued due to financial issues of the manufacturer. Nonetheless, the results reported above demonstrate the high biocompatibility of DLC. Additionally, in simulated body fluid, amorphous carbon has exhibited bio-inertness and a non-toxic response toward osteoblast-like cells [79,83,84,85,86,87]. This **suggests that** DLC has great potential as a superior substrate material for new bio-interfaces in terms of safety. Furthermore, it has been reported that coating 316 L stainless steel with DLC improves osteogenesis ability [6]. While it is not possible to detail all aspects of the biocompatibility of DLC and DLC-based materials here, it can be concluded that DLC is a promising coating material capable of providing antimicrobial properties while maintaining its functionality as a biomaterial.

## 10. Conclusions

Research on the medical applications of DLC as an antimicrobial coating is still in its early stages of development. DLC is a thin-film nanocarbon coating material known for its excellent biocompatibility and ability to confer antimicrobial properties to substrates. As described, various modifications can be applied to DLC, making it relatively easy to develop diverse properties compared to those of organic materials. The required antimicrobial performance varies depending on the substrate used in the application, the target microbial species, and the environment in which it is used. DLC has the potential to be a next-generation antimicrobial coating material that meets diverse needs.

Future research aimed at elucidating the reactions of bacteria, cells, and viruses at foreign material interfaces is expected to advance our understanding of the more intricate aspects of these interactions. The thickness of functional thin films has already entered the realm of molecular and atomic structures, with further thinning representing a challenge. As our understanding of the relationship between fine structures, surface potential, ζ-potential, and thin film interfaces advances, DLC’s potential for controlling biological responses is likely to attract increasing attention, thus representing a promising area for future development.

## Figures and Tables

**Figure 1 ijms-25-08593-f001:**
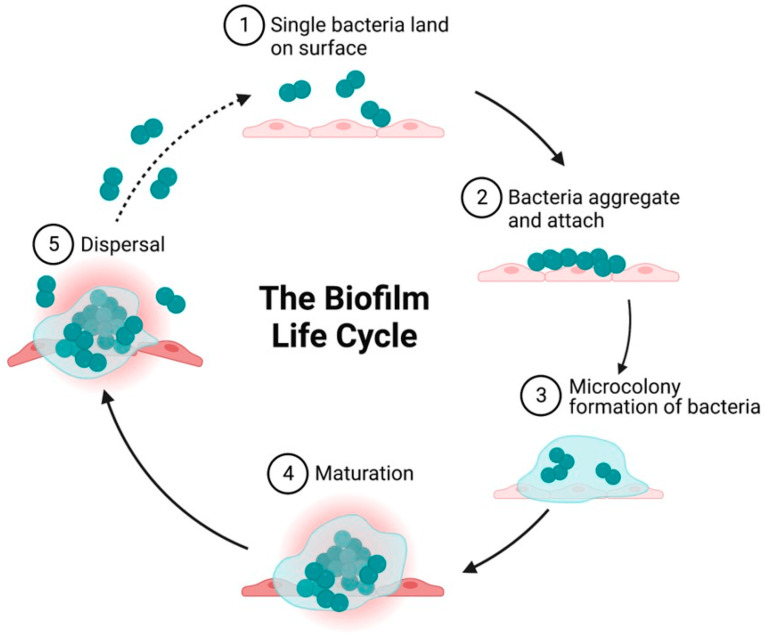
Schematic showing the growth cycle of a biofilm formed by a single bacterial species on a solid surface. (1) Reversible attachment of single planktonic bacteria to the surface. The first attach-ment of bacteria is influenced by attractive or repulsive forces generated by nutrient levels, pH, and surface temperature. (2) Aggregation of bacteria and irreversible attachment to surfaces. (3) Formation of an external matrix of multilayered complex biomolecules, microcolony formation, and EPS secretion that constitute the external matrix. Secretion of polysaccharides by biofilm-forming strains enables aggregation, adherence, and surface tolerance, allowing for improved surface colonization. (4) Maturation of biofilms and acquisition of a three-dimensional structure as they reach maturity. These three-dimensional structures rest on self-produced extra-cellular matrix components. (5) Fully mature biofilms detach, which allows bacterial cells to take on a planktonic state once again, thereby establishing biofilms in other locations (reprinted from [20]).

**Figure 2 ijms-25-08593-f002:**
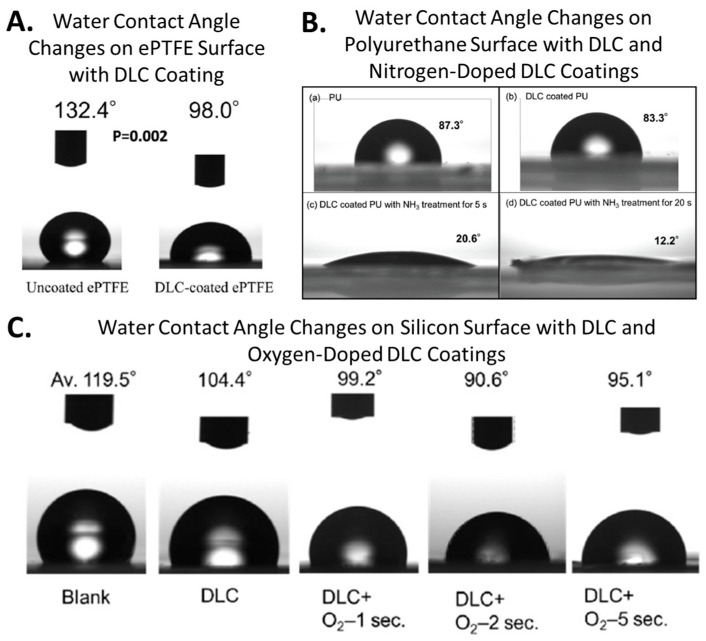
Examples of water contact angle (free surface energy) control with DLC. (**A**) Water contact angle changes on ePTFE surface with DLC coating (reprinted from [5]) (**B**); water contact angle changes on polyurethane surface with DLC and nitrogen-doped DLC coatings (reprinted from [13]) (**C**); water contact angle changes on silicon surface with DLC and oxygen-doped DLC coatings (reprinted from [11]).

**Figure 3 ijms-25-08593-f003:**
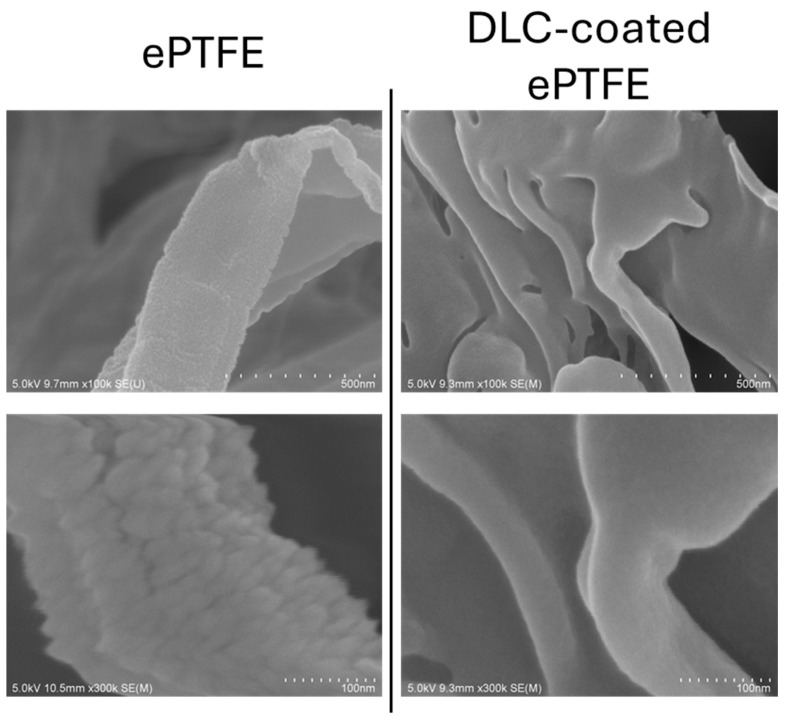
Surface smoothing of fibrous ePTFE structures using DLC coating. Application of DLC on the ePTFE surface clearly results in smoothing of each individual fiber within the fibrous structure of the ePTFE.

**Figure 4 ijms-25-08593-f004:**
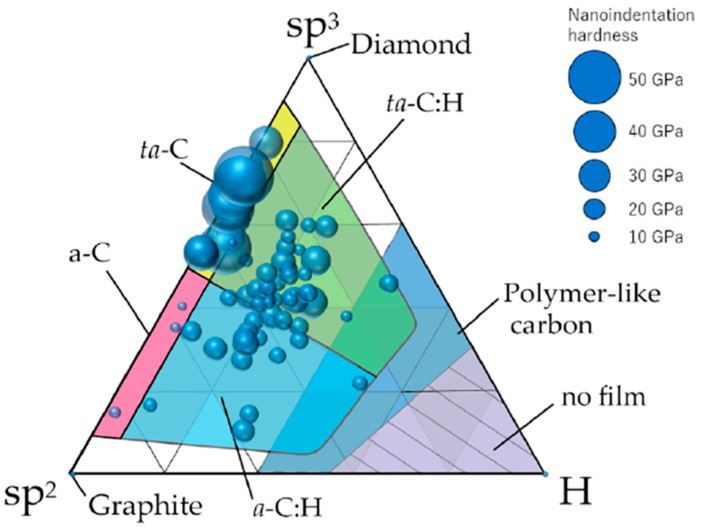
The distribution of 74 types of amorphous carbon films on a ternary diagram based on sp^2^ bonds, sp^3^ bonds, and hydrogen content. The diameter of each circle corresponds to the nanoindentation hardness of each amorphous carbon film. (Reprinted from [31]).

**Figure 5 ijms-25-08593-f005:**
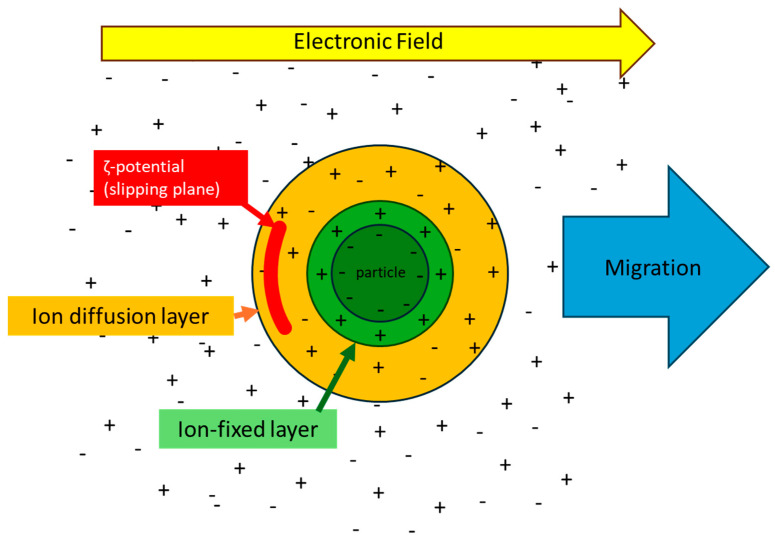
Image of ζ potential.

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
