# Peer review of "Development of Antimicrobial Surfaces Using Diamond-like Carbon or Diamond-like Carbon-Based Coatings"

_ijms, 2024, doi:10.3390/ijms25168593_

Round 1
Reviewer 1 Report
Comments and Suggestions for Authors
In this review manuscript, diamond-like carbon material was presented as a surface-coating antimicrobial material for implanted medical devices.
It was shown that surface modification and characterization are very important for gaining insight into antimicrobial properties. A discussion of DLC-based materials gives insight into the development of antimicrobial surfaces.
In this review manuscript, it was shown that DLC materials have the potential to be the next-generation coating material for medical devices.
The methodology was done well.
Conclusions are consistent with the results presented.
The manuscript is well-written, clear, and supported by relevant references. The recommendation is to publish the manuscript in its present form.
Author Response
Thank you for reviewing our paper. We are pleased to receive a good evaluation.
Reviewer 2 Report
Comments and Suggestions for Authors
This is an interesting manuscript, focusing on the development of antimicrobial surfaes based on diamond-like carbon. However, some problems could be identified:
-the title seems to not be in complete agreement with the manuscript content because title refers applications and the manuscript deals mainly with “DLC-based antimicrobial surface development” (this is in the last sentence of abstract).
-The novelty of this work should be better evidenced, including with the presentation and referencing the main reviews on proximal topics (example: https://www.mdpi.com/2079-6412/12/8/1088)
-the searching strategy should be presented (keywords, timespan, databases used,..)
-section 2 should be modified, in my opinion, because this is a review paper and not a book for study
-in my opinion, the Biocompatibility should be a chapter of the manuscript, given the high importance of this area
Comments on the Quality of English LanguageMinor editing of English language required
Author Response
We sincerely appreciate you taking the time out of your busy schedule to review our manuscript. Below, we have listed your questions along with our responses.
Comment: -the title seems to not be in complete agreement with the manuscript content because title refers applications and the manuscript deals mainly with “DLC-based antimicrobial surface development” (this is in the last sentence of abstract).
Answer: Thank you very much for your important comment. Indeed, you are correct. We have changed the title to “Development of Antimicrobial Surfaces using Diamond-like Carbon or Diamond-like Carbon based coatings”.
Comment: The novelty of this work should be better evidenced, including with the presentation and referencing the main reviews on proximal topics (example: https://www.mdpi.com/2079-6412/12/8/1088)
Answer: Thank you for your insightful comments. I am aware of the review you mentioned. This review is distinctly different from previous reviews related to DLC as it focuses specifically and in detail on the antimicrobial potential of DLC and DLC-based coating materials. While there are short comments in some reviews that mention the antimicrobial properties related to doping with elements like silver or associations with PTFE, none have investigated antimicrobial properties in such depth. Upon your suggestion, I re-examined PubMed and ScienceDirect but was unable to find any reviews with a similar concept. To highlight the uniqueness of this review, I added the statement, "To the best of our knowledge, this review represents the first comprehensive summary of Diamond-Like Carbon (DLC) and DLC-based antimicrobial coatings," at the end of the introduction.
Commen: the searching strategy should be presented (keywords, timespan, databases used,..)
Answer: There are relatively few publications related to the antimicrobial properties of Diamond-Like Carbon (DLC). A search on PubMed using the term "Diamond-like Coating" yields only around 700 results. Thus, a comprehensive search using only the keyword "diamond-like carbon" is sufficient to cover the relevant literature. We have examined publications available up to the end of April 2024. In addition to PubMed, we also used J-STAGE for our search. We have added the following statement to the introduction.
(Page 2, Line 75-79) For this review, literature searches were conducted using PubMed and J-STAGE. All relevant literature up to the end of April 2024 was included in the search. The search term "Diamond-like carbon" was used to identify relevant documents, focusing on both reviews related to DLC and research articles addressing the antimicrobial properties of DLC. This approach allowed us to collect comprehensive information on the subject.
Comment: section 2 should be modified, in my opinion, because this is a review paper and not a book for study
Answer: Thank you for your advice. I have revised Section 2 accordingly. Please review the revised manuscript. I hope the changes align with your intended corrections.
Comment: in my opinion, the Biocompatibility should be a chapter of the manuscript, given the high importance of this area
Answer: Thank you very much for your valuable feedback. We have added a new section, "9. Biocompatibility," where we discuss the biocompatibility of DLC and DLC-based coatings. We appreciate your review and look forward to your feedback.
Reviewer 3 Report
Comments and Suggestions for Authors
The review by Yasuhiro Fujii et al. focusses on a specific carbon material form known as diamond-like carbon (DLC), and its antimicrobial properties.
This review is a well-written article summing up the current knowledge on DLC and its application in antimicrobial surfaces, including biomedical scenarios.
Author Response
Thank you for taking the time in your busy schedule to review our manuscript. Below are your questions and our responses to them.
1) p4 l. 141-145: The authors are asked to explain more in detail the antimicrobial properties of DLC coatings, which they reported in the previous work on PU or silicone substrates. According to the Fig.2 A, the contact angle of DLC coated ePTFE is reduced to 98° (slightly hydrophobic), which is in the range of bacterial cell adhesion (see. L. 134-136), and the contact angle of DLC itself is given between 70 - 100° (see l. 140-141). This seems to be contradictory evidence.
Answer: Thank you for your important feedback. As we mentioned in the paragraph preceding the one you highlighted, there are multiple reports on the water contact angle that results in the worst bacterial adhesion. We believe this variability is due to differences in the substrate, type of DLC, surrounding environment, and bacterial species. We agree that the current structure made the explanation unclear, so we have provided the following additional explanation.
Page 4, Line 156: We have changed "approximately" to "aboutly." The reason for this change is that the relationship between water contact angle and bacterial adhesion is likely to vary significantly depending on the conditions, as mentioned above.
Page 4、Line159-163:
Pre: “Surfaces with intermediate contact angles are therefore likely to be disadvantageous for antimicrobial purposes.”
Post: “The specific water contact angle at which bacterial adhesion is maximized is likely to vary depending on the substrate, type of DLC, surrounding environment, and bacterial species. Therefore, a definitive threshold will be case-dependent. However, the observation that bacterial adhesion is most significant at intermediate water contact angles is an important finding.”
We have added a discussion regarding the polyurethane results from reference 7.
Page 4、Line 169-171: This is a comprehensive result of the increased smoothness provided by the DLC, the change in water contact angle from 90° to 74.5°, and the shift in zeta potential measured in a 10mM NaCl environment from approximately -4mV to around -11mV.
Additionally, we have included an explanation regarding the study on silicone tubes in an artificial urine environment (reference 8).
Page 4, line 173-180: The inner lumen of silicone tubes was coated with DLC, and artificial urine containing bacteria was circulated through these tubes for 14 days. The DLC coating inhibited the colonization of Pseudomonas aeruginosa and Escherichia coli on the inner surface of the silicone tubes and significantly reduced biofilm formation by Pseudomonas aeruginosa [8]. However, in this study, no antimicrobial effect against Staphylococcus aureus was observed under these conditions, suggesting that the antimicrobial performance of DLC may vary depending on the application context.
2) In Chapter 4. Tribology, the authors report on DLC-coated ePTFE microfibers and smoothing surface roughness of the fibers (as shown in Fig. 3). Is this the only report available examining the relation between tribology and DLC coating? Are there any quantitative results on the changed tribological properties of the coated surfaces? Is any relation known to antimicrobial properties, e.g. do bacterial cells grow on coated (or uncoated) ePTFE?
The authors are asked to address these points in more detail.
Answer: Thank you for your inquiry. The image presented as Figure 3 in our current work has not been previously published elsewhere, although we have reported similar images in Reference 5. For this instance, we have selected a different image from our collection. In Reference 5, we quantified the changes in smoothness by calculating the arithmetic mean roughness (Ra) and root mean square roughness (Rq). In that report, we demonstrated that DLC coating on the ePTFE surface resulted in a 65% reduction in Ra from 2.827 to 0.954 and a 68% reduction in Rq from 3.584 to 1.150. Similarly, for polyurethane surfaces, we reported that DLC coating reduced Ra by 41% and Rq by 45% (Reference 7). On silicon surfaces, we found that DLC coating resulted in a 70% reduction in Ra (Reference 8). Regarding the relationship between antimicrobial properties and smoothness, specifically the adhesion of bacteria and biofilm formation, we revisited the literature. It appears that isolating the effect of smoothness alone on bacterial adhesion and biofilm formation is challenging due to the difficulty in setting conditions that solely vary smoothness. Consequently, to our knowledge, no studies have exclusively examined the impact of smoothness on bacterial activity, such as adhesion and biofilm formation. We have added a few comments on this aspect as well. Based on the above, we have made the following changes.
(Page 5, Line 205-213) : We have conducted research on DLC (diamond-like carbon) coatings on resin substrates and confirmed that DLC coatings enhance surface smoothness on resins similarly to metals. In ePTFE, DLC coating resulted in a 65% reduction in arithmetic mean roughness (Ra) and a 68% reduction in root mean square roughness (Rq) [5]. Similarly, for polyurethane substrates, DLC coating reduced the surface Ra by 70% and Rq by 45% [7], and on silicon surfaces, DLC coating decreased Ra by 70% [8]. Although these values are likely influenced by the original surface microstructure and slight compositional differences of the substrates used, it is believed that the smoothness enhancement effect of DLC coating is also applicable to resin materials.
(Page 6, Line 226-229): It is generally accepted that large surface areas with rough textures promote bacterial adhesion. However, due to the difficulty of establishing conditions that vary only in smoothness, quantifying the extent to which smoothness affects bacterial activity or identifying specific topographic patterns that promote adhesion remains challenging.
3) In Chapter 5, the authors shortly report on hydrogen-rich DLC coatings and a possible relation to antimicrobial properties, however, it remains unclear why a higher hydrogen atom content in DLC material should lead to antimicrobial properties pe se. The H atoms are tightly bond in the chemical/physical structure of the DLC material, thus are not easily accessible. There should be literature available that discussed the effect of hydrogen atoms in highly structured, crystalline-like materials in opposite to easy-accessible or even free hydrogen in relation to antimicrobial properties of materials. Could the reported effect also be related to surface hardness effects of the DLC material?
Answer: Thank you very much for your important observations. As a general principle, it is the sterilizing action of ROS and hydrogen peroxide that predominantly plays a role, rather than that of hydrogen itself. It is true that hydrogen, when tightly bound to the substrate, appears to be inactive. However, the process by which hydrogen is incorporated into DLC is complex, with both tightly chemically bonded hydrogen and non-bonded hydrogen being present. Additionally, the distribution of non-bonded hydrogen appears to vary, and it remains uncertain how much biological activity hydrogen within the DLC possesses. Considering this, I have added references and included additional descriptions.
Regarding the relationship between surface hardness and antibacterial properties, I conducted investigation but was unable to find any significant insights. It is possible that biofilm formation is necessary for adhesion and growth on hard surfaces. Thus, the ability or inability to form a biofilm might influence the ease with which bacteria adhere to hard surfaces. However, bacteria that do not form biofilms can coexist within biofilms formed by other bacteria, so hardness does not necessarily correlate with antibacterial properties. Therefore, it remains unclear whether surface hardness impacts antibacterial effectiveness. Given the lack of notable findings on this matter, I decided not to include any specific comments on this topic.
Based on the above, I have add some comments as below.
(Page7, Line 258-277) These reactions are unlikely to occur when hydrogen and carbon are tightly chemically bonded, as in diamond-like carbon (DLC). However, the process of growing hydro-gen-containing DLC is highly complex [53,54], leading to variability in the modes of hy-drogen presence. Moreover, within DLC, hydrogen exists in two forms: hydrogen that is tightly chemically bonded to carbon, and hydrogen that is present but not tightly bonded, effectively trapped within the carbon matrix. It is suggested that there are regions within the DLC where trapped hydrogen accumulates, resulting in variations in hydrogen con-centration [55]. While loosely bound hydrogen is speculated to have potential antimicro-bial activity, no studies have been found to verify this. Detailed research is needed on the specifics of hydrogen presence, the ratio of tightly bonded to loosely bonded hydrogen, and the biological activity of loosely bonded hydrogen. Nevertheless, the DLC developed for our resin interior has a high hydrogen content of 30%, and under certain conditions, we have confirmed its antimicrobial properties [7, 8]. It is intriguing to consider whether the hydrogen within the DLC contributes to this effect. Additionally, several studies have reported the antimicrobial properties of DLC films. Levon et al. reported that DLC induced lower adherence and colony formation by Staphylococcus aureus than tantalum, titanium, and chromium [56]. Del Prado et al. also demonstrated the antibacterial effectiveness of DLC-coated ultra-high-molecular-weight polyethylene against nine different staphylo-coccal species [57]. Further research is required to elucidate the relationship between the structure of DLC and its antibacterial properties.
4) In Chapter 5, the authors further shortly discuss the antimicrobial performance of DLC doped material. However, the mentioned doping agents are well known to be cell-active, e.g. by cell-toxic properties of metals as discussed in l. 236-238. Thus, the authors are asked to supply more information on if the reported antibacterial performance is mainly an effect of the cell toxicity of the doping agents.
Answer: As you rightly pointed out, the antibacterial properties observed when metals or halogens are doped are primarily due to the cytotoxicity of the doped atoms. To emphasize the issue of cytotoxicity, I have changed statement about adverse effects of doped DLC as follows
(Page 7, Line 294-304) : However, the antibacterial mechanism observed when atoms are doped is primarily due to the cytotoxicity of the doped atoms. Therefore, careful consideration of the side effects related to cytotoxicity is necessary before practical application. Notably, many metal at-oms exhibit strong cytotoxicity when ionized [63,68]. Additionally, unexpected side effects that cause serious harm to biological systems may also arise beyond cytotoxicity. For in-stance, we observed that oxygen doping significantly reduces the blood compatibility of DLC surfaces (unpublished data), which were originally comparable to that of ePTFE vascular grafts [5]. Consequently, oxygen-doped DLC cannot be used for blood-contacting applications such as intravascular stents, artificial blood vessels, or vascular catheters. It is important to always bear in mind that doping atoms can significantly alter the properties of DLC.
5) In Chapter 6), as for surface free energy, what is the surface zeta potential of a DLC coating material alone, or on various substrates? It also remains unclear how it contributes to lower the potential of a given polymer surface? According to the previous Chapter 5., a DLC coating should be essentially be charge neutral, since there are no ions incorporated. The authors are asked to give more details/explain the reported effect.
Answer: The zeta potential is an intrinsic property of the material. However, since different temperatures, pH levels, and ionic strengths yield different values, it is essential to measure it under consistent conditions. Theoretically, a completely DLC-coated surface should exhibit a specific zeta potential value, but we have observed variations depending on the substrate. We hypothesize that this is due to the presence of pinholes in the DLC coating, albeit small in area. Since DLC is deposited via vapor deposition, clusters form in the plasma during deposition, sometimes detaching and creating gaps that form pinholes. We suspect that these pinholes allow the substrate to influence the zeta potential. It is particularly likely that thin DLC coatings are more susceptible to the influence of pinholes. As there is currently no technology to completely eliminate pinholes in DLC, variations in potential arise due to differences in substrate material. Additionally, hydrogen-containing DLC is not neutral; the DLC surface is terminated with hydrogen. Furthermore, DLC deposited in a vacuum chamber can incorporate oxygen due to residual oxygen pressure, leading to the formation of carboxyl groups, which can result in a negative potential. Considering these factors, we have added the following amendments to Chapter 6.
(Page 9, Line 351, Page 10, Line 365) Theoretically, a completely DLC-coated surface should exhibit a consistent ζ-potential value regardless of the substrate because ζ-potential is considered an intrinsic property of materials. However, we currently observe variations in ζ-potential depending on the sub-strate. We hypothesize that this is due to the presence of pinholes in the DLC coating, though small in area, through which the substrate influences the ζ-potential of the DLC surface. Since DLC is deposited via vapor deposition, clusters form in the plasma during deposition, sometimes detaching and leaving gaps that form pinholes. It is particularly likely that thin DLC coatings are more susceptible to the influence of pinholes. As there is currently no technology to completely eliminate pinholes in DLC, variations in potential arise due to differences in substrate material. Moreover, hydrogen-containing DLC is not neutral; the surface of DLC is terminated with hydrogen. Additionally, DLC deposited in a vacuum chamber can incorporate oxygen due to residual oxygen pressure, leading to the formation of carboxyl groups from carbon, hydrogen, and oxygen. Therefore, the ζ- poten-tial of DLC can vary widely. Further research is needed to measure and interpret the ζ-potential of DLC comprehensively.
6) In Chapter 8, it remains unclear why surface diffusion should be relevant for antimicrobial properties of DLC material? The authors speculate on low given (or available) evidence, and are asked to significantly improve this section, or delete it otherwise.
Answer: We are conducting research with the goal of developing materials with antibacterial properties and biomimetic characteristics for medical devices using DLC and DLC-based materials. It is likely that very few medical researchers are familiar with the concept of surface diffusion. To introduce this concept, we have dedicated a chapter in this paper.
DLC, composed solely of carbon, is known for its high hardness and stable, low-reactivity properties, which make surface diffusion unlikely. However, in cases where DLC incorporates hydrogen or oxygen atoms, or when various atoms are doped into the DLC, surface diffusion may occur. Additionally, the use of DLC at high temperatures can also lead to surface diffusion, potentially affecting the surface characteristics and mechanical properties.
Since it is rare to use pure carbon DLC, understanding the properties of DLC and considering this phenomenon when selecting appropriate applications and conditions is crucial. Especially in biomedical applications, the impact of surface diffusion on biological responses should be recognized and incorporated into material design. Therefore, we have included this chapter to emphasize its importance. In light of this, we have completely rewritten this section and added a reference (Ref 79).
Round 2
Reviewer 2 Report
Comments and Suggestions for Authors
The document as improved and now it is more acceptable for publication.
Comments on the Quality of English LanguageMinor editing of English language required